# Effects of Coupling Action of Load and Temperature on the Lubricity of Coke Powder

**Jin Xiang \*, Wenan Cai and Haidong Zhang**

Department of Mechanics, Jinzhong University, Jinzhong 030619, China
* Correspondence: xiangjin7569989@163.com

**Abstract:** By introducing the coke powder into the friction pair, the effects of load and temperature on the lubricity of coke powder at the frictional interface are studied. Before the test, the microcrystalline structures of coke powder at different temperatures are characterized by X-ray diffraction (XRD). After the test, the friction surfaces are observed by electron microscope and energy spectrum, the oxide compositions of friction surface are characterized by XRD, and the lubrication mechanism of coke powder is investigated. The results show that at RT (room temperature), as the load increases, the forming of the powder layer gradually deteriorates. When the load is 5 MPa, at a low temperature, the powder layer is thicker, and the coke powder exhibits better lubricity. In contrast, at a high temperature, affected by the temperature, the lubricity of coke powder declines, and the friction coefficient is higher.

**Keywords:** load; temperature; coke powder; powder layer lubrication





## 1. Introduction

Coke pushing is a very important process in coke oven production [1]. The coke pusher must enter the high-temperature carbonization chamber to push coke. Due to the existence of coke powder, the friction interface is formed at the bottom of the coke pusher, coke powder, and ground of the carbonization chamber (refractory brick). Due to the tribological characteristics of the interface directly affecting the stability of coke pushing, in this paper, the effects of load and temperature on the lubricity of coke powder at the frictional interface are studied.

Solid lubricants are widely used in many industrial fields [2]. In addition, the matrix forms composite materials by adding solid lubricants, which have excellent lubrication performance [3,4]. Wang et al. [5] introduced graphite powder into the friction pair to study the macro and micro characteristics of the warm formed friction interface powder layer. The results showed that the friction coefficient first increased, then decreased, and then increased slightly with the increase in temperature. The temperature had a great influence on the friction coefficient of the interface, the state of the powder layer, and the lubrication performance of graphite powder. Chang et al. [6] studied the effects of interface temperature and contacted load on the heat transfer coefficient of copper alloy, graphite powder, aluminum oxide powder, and steel. The results show that the temperature could cause changes in the thermal resistance and oxide layer thickness of solid lubricant, while load could cause changes in the actual contact area and contact properties of the interface. Shi et al. [7] prepared NiAl matrix self-lubricating composites with different solid lubricant additions (PbO, $Ti_3SiC_2$-$MoS_2$, $Ti_3SiC_2$-$WS_2$) by spark plasma sintering (SPS). The results showed that $MoS_2$ exhibited better lubrication at low temperatures, $Ti_3SiC_2$ exhibited better lubrication at high temperatures, while $Ti_3SiC_2$-$MoS_2$ binary lubricant exhibited the best synergetic lubrication.

Xiang et al. [8] compared the interfacial friction characteristics of non-lubrication and coke powder lubrication by experiments and found that coke powder played a very

important role in lubrication. They also studied the effects of velocity and load on the lubrication performance of coke powder at room temperature. However, research on the tribological properties of coke powder under the coupling of load and temperature has not been reported. Firstly, the microcrystalline structures of coke powder at different temperatures are studied by XRD. In addition, the effects of load on the lubricity of coke powder at RT are analyzed, and the load at the optimal lubricity of coke powder is obtained; on this basis, the effects of temperature on the lubricity of coke powder are further analyzed, thus providing reliable operating parameters to ensure the stability of coke pushing.

## 2. Experimental

### 2.1. Experimental Material

The upper sample was a cylinder with a diameter of 5 mm and a height of 10 mm, made of steel. The lower sample with a dimension of 30 mm ×30 mm ×10 mm was made of refractory brick, with a surface roughness Ra of 12.68 micron. Before the tests, the coke powder was applied to the lower sample, and the coke powder used in the tests was 50 micron with the amount of 3 g for each test. Friction test diagram is shown in Figure 1.

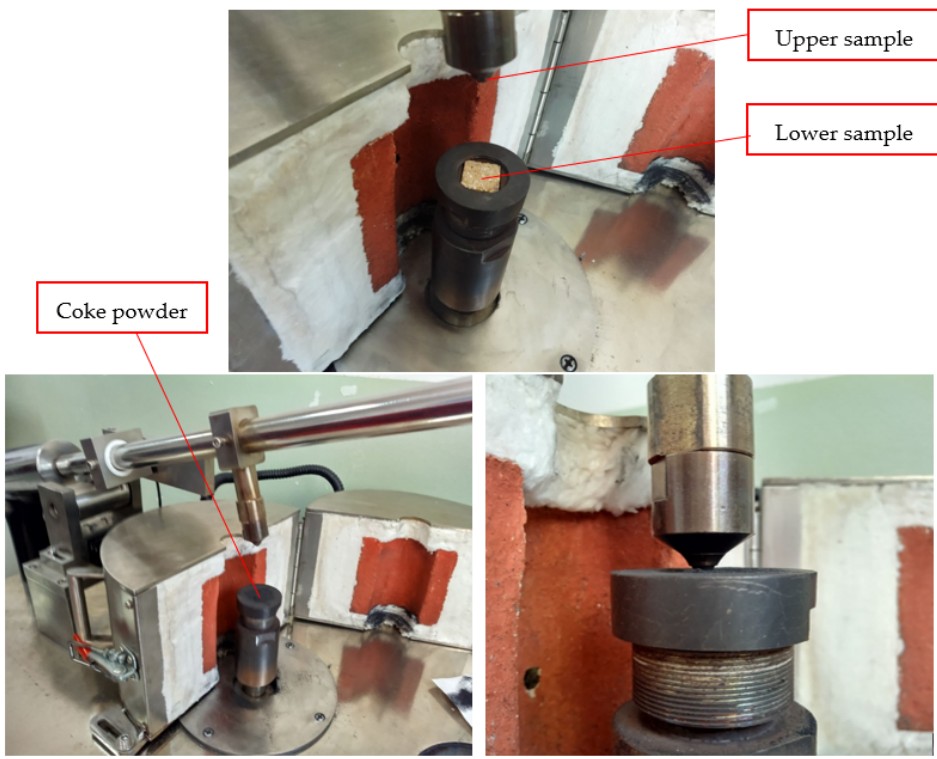

**Figure 1.** Friction test diagram.

### 2.2. Experimental Method

Tribological tests were performed using a high-temperature tribometer (HT-4001, Huijin Tieer, Hangzhou, China). During the tests, the upper sample was fixed, and the lower sample was rotated. The test loads were 5 MPa, 10 MPa, 15 MPa, and 20 MPa, respectively. The sliding velocity of the lower sample was 0.45 m/s, the test temperature was room temperature (RT), 400 °C, and 600 °C, respectively, and the test time was 150 s. Each experiment was tested three times, and the friction coefficient was taken as the average of three values.

### 2.3. Characterization

XRD diffraction (JEOL-2100F, JEOL, Tokyo, Japan) analysis was tested using a Cu target, with a voltage of 40 kV, a current of 100 mA, and a scanning velocity of $1.0°$ $min^{-1}$ for 2θ values between $20°$ and $80°$. The surface morphology of samples was characterized

by scanning electron microscopy (SEM) (JSM 6390, JEOL Tokyo, Japan) equipped with energy dispersive X-ray spectrometry (EDS) (JSM 6390, JEOL Tokyo, Japan)

## 3. Results and Discussion

### 3.1. Microcrystalline Structure of Coke Powder at Different Temperatures

Before the experiments, the microcrystalline structures of coke powder at different temperatures are analyzed by XRD, shown in Figure 2. It can be seen that there are (002) and (100) diffraction peaks, while there is a (004) diffraction peak in Figure 2b,c. The more characteristic peaks, the higher the graphitization degree of the coke powder.

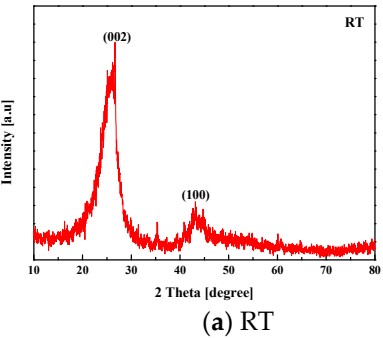

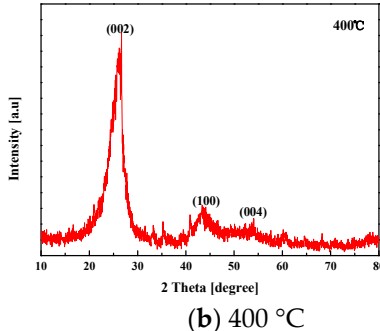

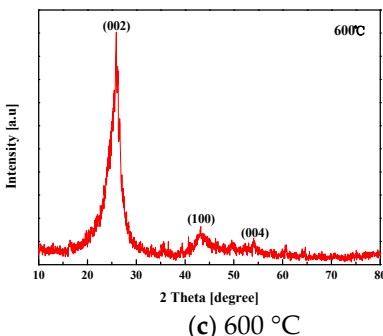

(**a**) RT          (**b**) 400 °C          (**c**) 600 °C

**Figure 2.** XRD patterns of coke powder at different temperatures.

Graphitization degrees of coke powder at RT, 400 °C, and 600 °C are calculated using the Bragg equation [9], Scherrer equation, and Mering—Maire equation [10,11].

$$d_{002} = \frac{\lambda}{2\sin\theta_{002}} \tag{1}$$

$$L_c = \frac{0.89\lambda}{\beta\cos\theta_{002}} \tag{2}$$

$$g = \frac{0.3440 - d_{002}}{0.3440 - 0.3354} \tag{3}$$

where $d_{002}$ is the interlayer spacing and $\lambda$ is the X-ray wavelength (0.14506 nm). $L_c$ is the average crystalline thickness, $\beta$ is the full width at half maximum intensity of (002) diffraction peak, $2\theta$ is the peak position, and g is the graphitization degree.

The graphitization degree of coke powder at RT, 400 °C, and 600 °C are 64.19%, 70.47%, and 73.02%, respectively. This indicates that the higher the temperature, the more the aromatic layer structure of carbon atoms of coke powder tends to the aromatic layer structure of carbon atoms of graphite, the graphitization degree of coke powder gradually increases, and it may be inferred that the lubrication performance of coke powder improves with the increase of temperature.

### 3.2. Effects of Load on the Friction Coefficient and Powder Layer at RT

3.2.1. Friction Coefficient

As shown in Figure 3, the friction coefficients increase with the increase in load. Its mechanism is that the larger load is not conducive to the formation of a powder layer, and the lubricity of coke powder declined, increasing friction coefficient.

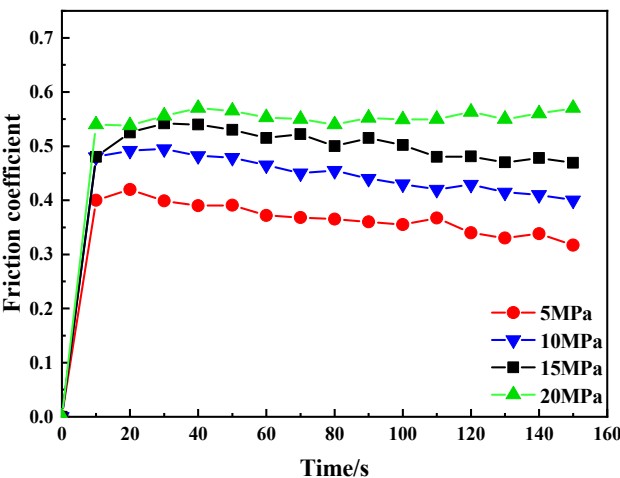

**Figure 3.** Friction coefficient at different loads.

3.2.2. Surface Morphology and Element Content of Powder Layer

As shown in Figure 4 and Table 1, the surfaces of the upper and lower samples are isolated by coke powder. Due to shear action, the coke powder adheres to the surface of the upper sample, forming a powder layer.

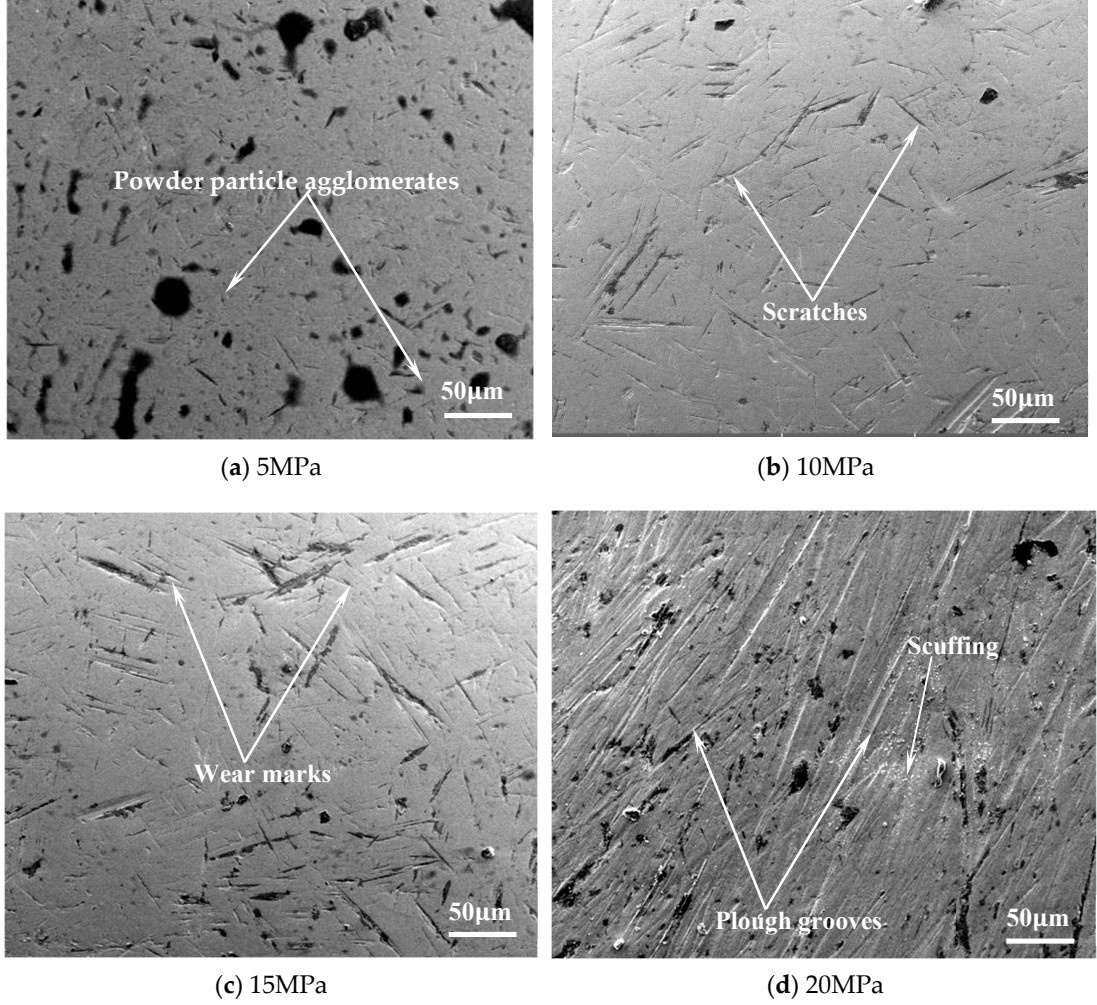

**Figure 4.** SEM morphology of powder layer at different loads.

**Table 1.** Element analysis of powder layer by EDS (ωt/%).

| Load | C | Fe | Si | Total |
|---|---|---|---|---|
| 5 MPa | 19.15 | 80.68 | 0.17 | 100.00 |
| 10 MPa | 12.43 | 87.42 | 0.15 | 100.00 |
| 15 MPa | 4.56 | 95.20 | 0.24 | 100.00 |
| 20 MPa | 0.85 | 98.78 | 0.37 | 100.00 |

When the load is 5 MPa, the content of element C is 19.15%, forming a relatively dense and complete powder layer. Shear action mainly occurs inside the powder layer [12,13], and there are agglomerates of coke powder in local positions, so the friction coefficient is lower.

When the load is 10 MPa, the content of element C is 12.43%. Shear action still mainly occurs inside the powder layer; however, the thickness of the powder layer is less than that of 5 MPa, and the lubricity of coke powder is declined. There are some scratches on the surface, which are formed by the action of hard abrasive particles.

When the load is 15 MPa, the content of element C is 4.56%, forming a thin powder layer. Shear action mainly occurs between the powder layer and the surfaces of samples or inside the powder layer. Due to the thin powder layer and limited lubricity of coke powder, there are many deep strip-shaped wear marks on the surface, and the friction coefficient is higher.

When the load is 20 MPa and element C's content is only 0.85%, shear action mainly occurs between the upper and lower samples. There are many plow grooves, scuffing occurs locally, surface wear is relatively serious, and the friction coefficient is the highest.

It can be found that with the increase of load, the powder layer gradually becomes thinner, and the load is 5 MPa; the coke powder exhibits the best lubrication state.

### 3.3. Effects of Temperature on Friction Coefficient and Powder Layer at 5 MPa

#### 3.3.1. Friction Coefficient

As shown in Figure 5, at 5 MPa, the friction coefficient at 600 °C is higher than that at RT, and the friction coefficient at 400 °C reaches the maximum.

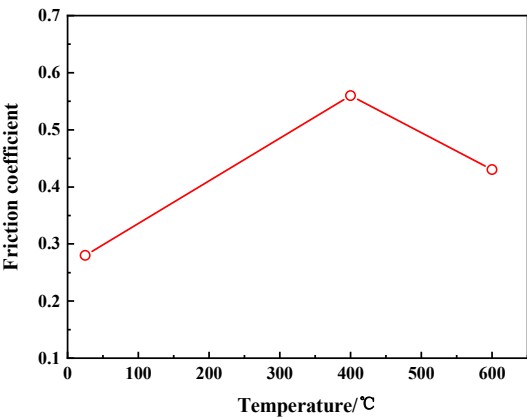

**Figure 5.** Friction coefficient at different temperatures.

#### 3.3.2. SEM Morphology and Elemental Content of Powder Layer

As shown in Figure 6 and Table 2, a layer of powder is formed on the friction surface.

At RT, the content of element C is 19.15%, forming a compact and thicker powder layer. The friction coefficient at RT is lower because the shear action occurs mainly inside the powder layer, and the powder layer is thicker, so the coke powder exhibits better lubricity. In addition, the hardness of the upper sample is higher, which is conducive to the formation of a powder layer.

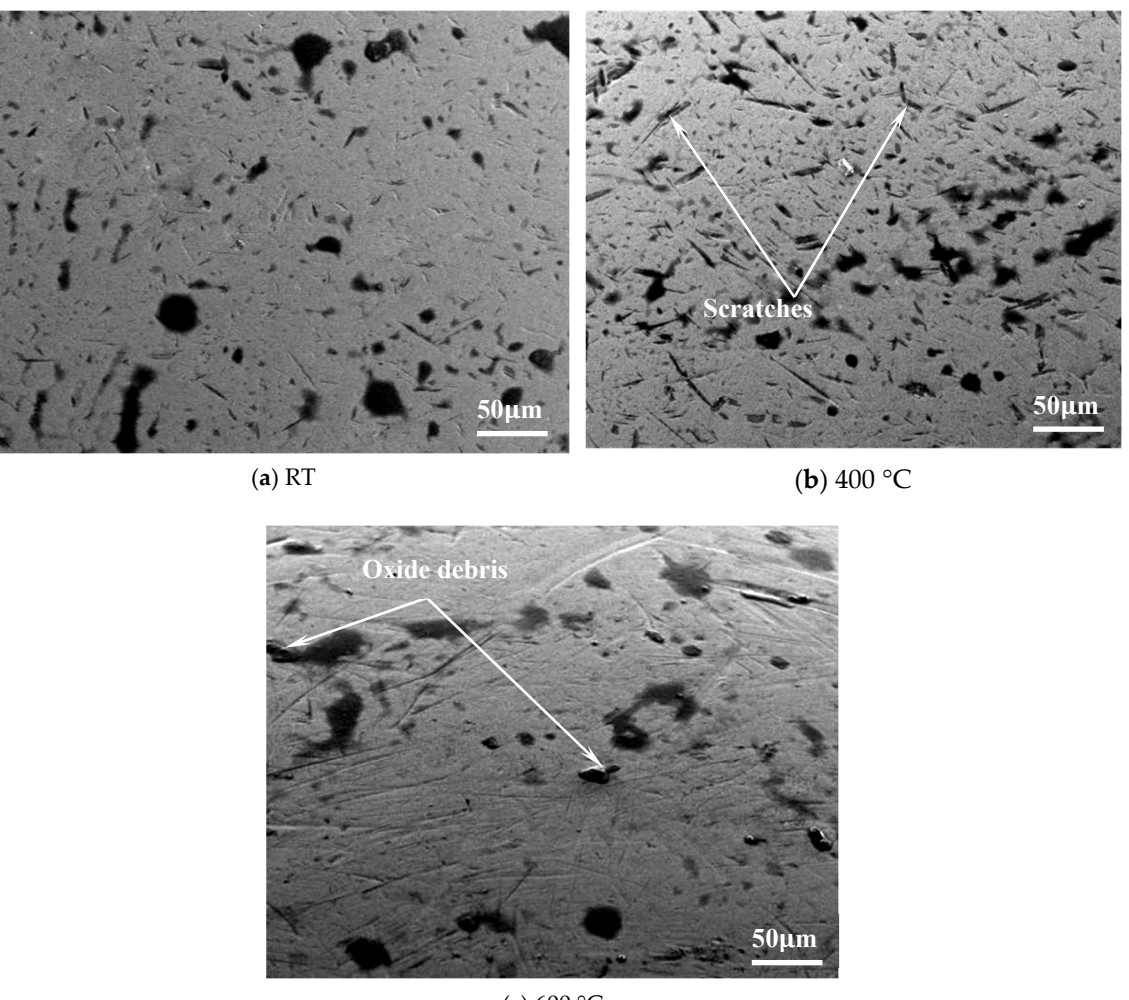

**Figure 6.** SEM morphology of powder layer at different temperatures.

**Table 2.** Element analysis of powder layer by EDS (ωt/%).

| Temperature | C | Fe | Si | O | Total |
|---|---|---|---|---|---|
| 20 °C | 19.15 | 80.68 | 0.17 | 0.00 | 100.00 |
| 400 °C | 10.34 | 85.25 | 1.42 | 2.99 | 100.00 |
| 600 °C | 7.87 | 85.20 | 0.11 | 6.82 | 100.00 |

At 400 °C, the content of element O is 2.99%, the oxidation of friction surface is not obvious, the amount of element C is 10.34%, and the effects of abrasive hard particles result in scratches on the surface of the powder layer. Although the shear action still occurs mainly inside the powder layer, the friction coefficient is the highest. This may be referred to as the precipitation of dis-lubricious silica ($SiO_2$) in the powder layer as analyzed by EDS (as shown in Table 2), which works against the lubricity of coke powder [14]. Despite the better lubricity of coke powder at 400 °C, due to the effect of $SiO_2$, the lubricity declines, and the friction coefficient reaches the maximum.

At 600 °C, the content of element O reaches 6.82%, the inner layer of the friction surface is covered with oxide film, and the outer layer is covered with the powder layer; the content of element C is 7.87%. The oxide film mainly includes $Fe_2O_3$, $Fe_3O_4$, and FeO (shown in Figure 7). Different oxides lead to weak bonding between oxides and are easily damaged, so the overall hardness is low [15]. Due to the formation of oxide film, the coke powder interacts with the oxide film. Due to the low hardness of the oxide film, on the one hand, the shearing of friction interface mainly occurs between the oxide film and the powder

layer or inside the powder layer, resulting in the decline of lubricity of coke powder, on the other hand, the adhesion of coke powder is weakened, the powder layer is thinner, and the coke powder lubrication is declined. In addition, the oxide film is a brittle phase, which produces some large-scale oxide debris (as shown in Table 3) during the shearing with the coke powder, which is also not conducive to coke powder lubrication [16]. Despite the best lubrication performance of coke powder at 600 °C, the lubricity declined due to the oxide film, so the friction coefficient is higher.

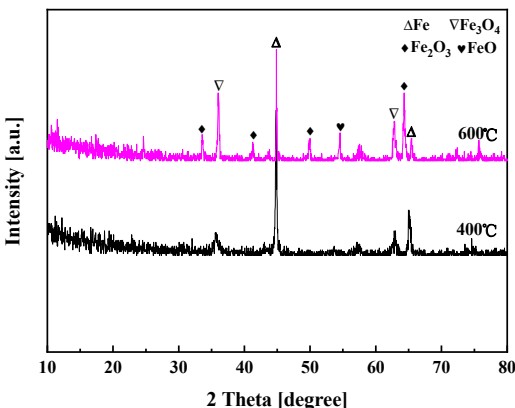

**Figure 7.** XRD patterns of friction surface.

**Table 3.** Element analysis of oxide debris by EDS (ωt/%).

| Temperature | C | O | Fe | Total |
|---|---|---|---|---|
| 600 °C | 1.13 | 46.6 | 52.27 | 100.00 |

## 4. Conclusions

In order to provide reliable operating parameters to ensure the stability of coke pushing, this study has attempted to analyze the coupling effect of load and temperature on the friction characteristics of coke powder lubrication. Some conclusions can be as follows:

1. At 0.45 m/s, with the increase in load, the powder layer gradually becomes thinner, indicating that a larger load is not conducive to coke powder lubrication.
2. When the load is 5 MPa, at RT, the powder layer is thick, and the coke powder exhibits better lubricity. At 400 °C, the thickness of the powder layer is thin. Moreover, the precipitation of dis-lubricious silica ($SiO_2$) in the powder layer leads to the maximum friction coefficient. At 600 °C, under the effect of oxide film, the powder layer is the thinnest, and the lubricity of coke powder declined.

**Author Contributions:** Conceptualization, J.X.; methodology, J.X. and W.C.; validation, J.X. and H.Z.; writing—original draft preparation, J.X.; writing—review and editing, W.C.; visualization, H.Z.; supervision, J.X.; project administration, J.X.; funding acquisition, J.X. All authors have read and agreed to the published version of the manuscript.

**Funding:** This research was funded by Fundamental Research Program of Shanxi Province, China; grant number No. 202203021222288.

**Institutional Review Board Statement:** Not applicable.

**Informed Consent Statement:** Not applicable.

**Data Availability Statement:** Not applicable.

**Acknowledgments:** All individuals included in this section have consented to the acknowledgement.

**Conflicts of Interest:** The authors declare no conflict of interest.

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
