# Peer review of "Effects of Coupling Action of Load and Temperature on the Lubricity of Coke Powder"

_coatings, doi:10.3390/coatings13050939_

Round 1

Reviewer 1 Report

In this study, the authors have characterized the microcrystalline structure of coke powder at different temperatures and the oxide composition of friction surface by XRD technique. Then, they investigated the effect of load and temperature on the lubricity of coke powder at the frictional interface. Their results showed that the formation of powder layer is less favorable with increasing the load values. Furthermore, at low temperature and at a load of 5MPa, the powder layer is thicker and the coke powder exhibits better lubricity, while the lubricity of coke powder decreases at high temperature. The motivation of the work and the approach adopted are well. The manuscript is publishable after the below mentioned questions/comments are addressed.

Comment:

- What is the effect of sliding velocity on friction coefficient and powder layer?

- Please bring the Bragg, Scherrer and Mering-Maire equations for calculating the graphitization degree of coke powder at different temperature and describe how the graphitization degrees of 64.19%, 70.47% and 73.02% are calculated based on these equations.

The authors use a combination of numerical simulation and experiment to study the change laws of the surface topography during the wear process. A time-varying wear calculation method is proposed, which includes a time-varying wear calculation model for worn surface and a compensation wear calculation model for unworn surface. The control variable method is used to analyze the influence of the rotational speed and the external load on the characteristic parameters of surface topography, which provides a new approach for solving tribological problems. There is a good agreement between the results of simulation and experiments. The results of this study can be found useful for the development and manufacture of friction pairs.

The motivation of the work and the approach adopted are well. The manuscript is publishable after the below mentioned questions/comments are addressed.

Comments

1) What is the effect of temperature on the wear process? How do your results change with temperature?

2) Please compare the accuracy of the model used in this study with other models reported in the literature for prediction of the properties. 

Author Response

Dear Editor:

     "Response to the reviewer’s comments", Please see the attachment.

Reviewer 2 Report

Journal: Coatings

Manuscript ID: coatings 2373242

Manuscript Type: Research Article

Manuscript Title: Effects of Coupling Action of Load and Temperature on the Lubricity of

Coke Powder  

Comments:

1.     Introduction: The authors mentioned that “the research on the tribological properties of coke powder under the coupling of load and temperature has not been reported”. However, the literature / background elaboration is very minimal to confirm the novelty of this work. As the background of this work presented in this manuscript is too brief. I would suggest the authors to bring in in-depth literature review to bridge the research gap between previous research and this work and to demonstrate the novelty of this work in this manuscript.

2.     Abstract: It seems no highlight or significant results presented from this work.

3.     Results and discussion: For Figure 1, the caption is not complete, missing with the indication of (a), (b), (c). Figure 1 (c) is not complete as the unit is missing. Also, it is difficult to compare the three different samples at three different temperature if the patterns are separated presented as the intensity are all presented in arbitrary unit (a.u.). This needs to be addressed and revised.

4.     Page 3 and Page 4: The figure numbers for the microstructures are missing. Pertaining to these microstructural images, for (a) 5 MPa, the labelling “Spot” seems incorrect and misleading. It seems to be a pore. As for (b) and (c), both labelling seems contradicting.

5.     The numbering of the figures is incorrect for Figure 4 and Figure 6.

6.     For the Figure presented in Page 6, i.e. the microstructural images, the Figure (b) seems incomplete. Also, can the authors confirm the “oxide debris”?

7.     Seems that the authors did not benchmark comprehensively the results from this work with others.

In general, the authors have attempted to study the effects of coupling action of load and temperature on the lubricity of coke powder. However, based on the evaluation above, this work seems not ready to be published yet. The authors should address all comments accordingly before resubmission. 

Author Response

Dear Editor:

     " Response to the reviewer’s comments ", please see the attachment.

Reviewer 3 Report

1. The authors should include the standards for experimental tests.

2. Subsection 3.1, Figure 1(c) should be rewritten as 6000C in the below of graph.

3. After subsection 3.1 coming subsection 3.3?. It should be checked.

4. Table 1, dot before load must be removed.

5. Conclusion, text font should be corrected.

6. I strongly recommend to check stile of presented references.

Minor editing of English language may help to improve the quality of manuscript. 

Author Response

(The authors gave the same response as above.)

Reviewer 4 Report

The paper describes an interesting phenomenon of the formation of a carbon layer from coke powder during friction, and investigates the effect of load and temperature on the friction coefficient. The topic of the study corresponds to the special issue chosen by the authors, but the text is of poor quality. It seems that this is a draft, and not a full-fledged paper. I advise you to submit the paper again after serious revision and additional research. I hope that the comments below will be useful in the revision.

1. A more detailed description of the experiment is needed. It is desirable to include a drawing or photograph of the friction scheme.

2. The experiments were carried out at a fixed speed. How was this speed chosen? Based on practical necessity?

3. How many repetitions of friction tests were there? Judging by the absence of any range of results, the points in Fig.2 are derived from the results of  single tests (for fixed loads and temperatures).

4. Why in Fig. 4 at room temperature is the COF equal to about 0.3? This is not a steady state or average value, but a single point at the end of the test.

5. Why is the counter-body not examined? Does it have carbon films on it? This is important for understanding friction mechanisms.

6. It would be useful to measure the layer thickness. If it is possible to make a cross-section, then one can also examine the structure of the layer.

7. Does the layer change over time? The answer can be given by experiments of various durations.

8. Abbreviations without decoding are used in the abstract.

9. There is no caption to Fig.3.

10. It is not written that in 3.3 results are given only for room temperature.

Lots of long sentences. In fact, they contain different phrases. This greatly complicates the understanding of the text.

Author Response

(The authors gave the same response as above.)

Round 2

Reviewer 2 Report

I did not find the response file from the authors specific to my queries. When I download the response file, the responses are addressing to different reviewers. Also, I still think that the "spot" in Figure 3 (a) is wrong. In addition, the captions are incomplete for Figure 3. In my opinion, the authors may need more time to prepare the manuscript before re-submitting to Coating journal. 

Author Response

Dear Editor:

     "Response to the reviewer’s comments", please see the attachment.

Reviewer 4 Report

The authors have improved the paper, but corrections are still needed:

1. Abbreviations in the abstract as they were, and remain.

2. The figure numbers are wrong.

3. On the curves it is necessary to indicate not only the average values of the coefficient of friction, but also the ranges for the three tests.

4. If there are no transfer films on the counter body, write about it, explaining the reason.

There are many erroneous wording and typos.

Author Response

(The authors gave the same response as above.)

Round 3

Reviewer 2 Report

The authors have addressed the comments accordingly.